# Brain-Like Object Recognition with High-Performing Shallow Recurrent ANNs

**Jonas Kubilius**[*,1,2], **Martin Schrimpf**[*,1,3,4],
**Kohitij Kar**[1,3,4], **Rishi Rajalingham**[1], **Ha Hong**[5], **Najib J. Majaj**[6], **Elias B. Issa**[7], **Pouya Bashivan**[1,3], **Jonathan Prescott-Roy**[1], **Kailyn Schmidt**[1], **Aran Nayebi**[8], **Daniel Bear**[9], **Daniel L. K. Yamins**[9,10], and **James J. DiCarlo**[1,3,4]

[1]McGovern Institute for Brain Research, MIT, Cambridge, MA 02139
[2]Brain and Cognition, KU Leuven, Leuven, Belgium
[3]Department of Brain and Cognitive Sciences, MIT, Cambridge, MA 02139
[4]Center for Brains, Minds and Machines, MIT, Cambridge, MA 02139
[5]Bay Labs Inc., San Francisco, CA 94102
[6]Center for Neural Science, New York University, New York, NY 10003
[7]Department of Neuroscience, Zuckerman Mind Brain Behavior Institute, Columbia University, New York, NY 10027
[8]Neurosciences PhD Program, Stanford University, Stanford, CA 94305
[9]Department of Psychology, Stanford University, Stanford, CA 94305
[10]Department of Computer Science, Stanford University, Stanford, CA 94305
[*]Equal contribution

## Abstract

Deep convolutional artificial neural networks (ANNs) are the leading class of candidate models of the mechanisms of visual processing in the primate ventral stream. While initially inspired by brain anatomy, over the past years, these ANNs have evolved from a simple eight-layer architecture in AlexNet to extremely deep and branching architectures, demonstrating increasingly better object categorization performance, yet bringing into question how brain-like they still are. In particular, typical deep models from the machine learning community are often hard to map onto the brain's anatomy due to their vast number of layers and missing biologically-important connections, such as recurrence. Here we demonstrate that better anatomical alignment to the brain and high performance on machine learning as well as neuroscience measures do not have to be in contradiction. We developed CORnet-S, a shallow ANN with four anatomically mapped areas and recurrent connectivity, guided by Brain-Score, a new large-scale composite of neural and behavioral benchmarks for quantifying the functional fidelity of models of the primate ventral visual stream. Despite being significantly shallower than most models, CORnet-S is the top model on Brain-Score and outperforms similarly compact models on ImageNet. Moreover, our extensive analyses of CORnet-S circuitry variants reveal that recurrence is the main predictive factor of both Brain-Score and ImageNet top-1 performance. Finally, we report that the temporal evolution of the CORnet-S "IT" neural population resembles the actual monkey IT population dynamics. Taken together, these results establish CORnet-S, a compact, recurrent ANN, as the current best model of the primate ventral visual stream.

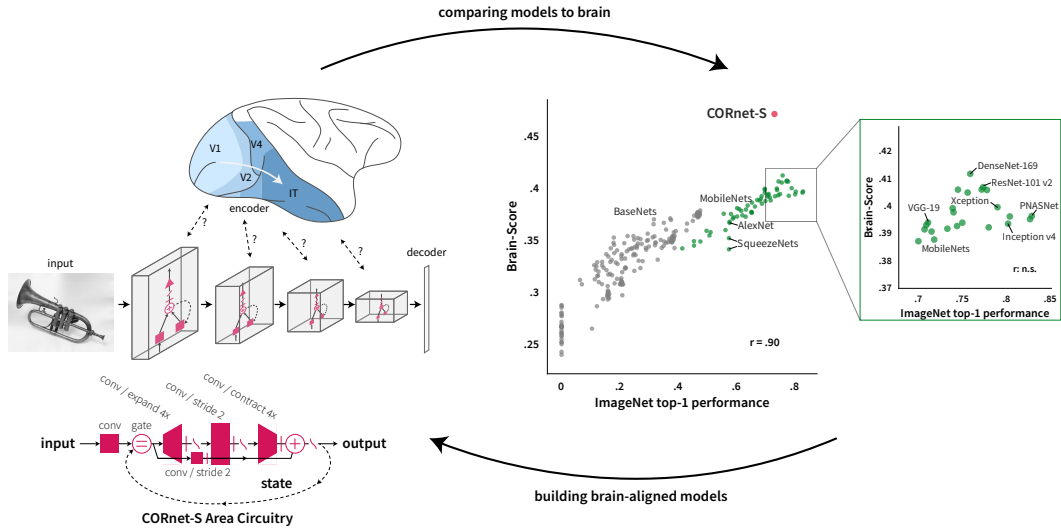

Figure 1: **Synergizing machine learning and neuroscience through Brain-Score (top).** By quantifying brain-likeness of models, we can compare models of the brain and use insights gained to inform the next generation of models. Green dots represent popular deep neural networks while gray dots correspond to various exemplary small-scale models (BaseNets) that demonstrate the relationship between ImageNet performance and Brain-Score on a wider range of performances (see Section 4.1). CORnet-S is the current best model on Brain-Score. **CORnet-S area circuitry (bottom left).** The model consists of four areas which are pre-mapped to cortical areas V1, V2, V4, and IT in the ventral stream. $V1_{COR}$ is feed-forward and acts as a pre-processor to reduce the input complexity. $V2_{COR}$, $V4_{COR}$ and $IT_{COR}$ are recurrent (within area). See Section 2.1 for details.

# 1 Introduction

Notorious for their superior performance in object recognition tasks, artificial neural networks (ANNs) have also witnessed a tremendous success in the neuroscience community as currently the best class of models of the neural mechanisms of visual processing. Surprisingly, after training deep feedforward ANNs to perform the standard ImageNet categorization task [6], intermediate layers in ANNs can partly account for how neurons in intermediate layers of the primate visual system will respond to any given image, even ones that the model has never seen before [46, 48, 17, 9, 4, 47]. Moreover, these networks also partly predict human and non-human primate object recognition performance and object similarity judgments [33, 21]. Having strong models of the brain opened up unexpected possibilities of noninvasive brain-machine interfaces where models are used to generate stimuli, optimized to elicit desired responses in primate visual system [2].

How can we push these models to capture brain processing even more stringently? Continued architectural optimization on ImageNet alone no longer seems like a viable option. Indeed, more recent and deeper ANNs have not been shown to further improve on measures of brain-likeness [33], even though their ImageNet performance has vastly increased [34]. Moreover, while the initial limited number of layers could easily be assigned to the different areas of the ventral stream, the link between the handful of ventral stream areas and several hundred layers in ResNet [10] or complex, branching structures in Inception and NASNet [41, 27] is not obvious. Finally, high-performing models for object recognition remain feedforward, whereas recent studies established an important functional involvement of recurrent processes in object recognition [42, 16].

We propose that aligning ANNs to neuroanatomy might lead to more compact, interpretable and, most importantly, functionally brain-like ANNs. To test this, we here demonstrate that a neuroanatomically more aligned ANN, *CORnet-S*, exhibits an improved match to measurements from the ventral stream while maintaining high performance on ImageNet. CORnet-S commits to a shallow recurrent anatomical structure of the ventral visual stream, and thus achieves a much more compact architecture while retaining a strong ImageNet top-1 performance of 73.1% and setting the new state-of-the-art in predicting neural firing rates and image-by-image human behavior on *Brain-Score*, a novel large-scale benchmark composed of neural recordings and behavioral measurements. We identify that these

results are primarily driven by recurrent connections, in line with our understanding of how the primate visual system processes visual information [42, 16]. In fact, comparing the high level ("IT") neural representations between recurrent steps in the model and time-varying primate IT recordings, we find that CORnet-S partly captures these neural response trajectories - the first model to do so on this neural benchmark.

## 2 CORnet-S: Brain-driven model architecture

We developed CORnet-S based on the following criteria (based on [20]):

(1) **Predictivity**, so that it is a mechanistic model of the brain. We are not only interested in having correct model outputs (behaviors) but also internals that match the brain's anatomical and functional constraints. We prefer ANNs because neurons are the units of online information transmission and models without neurons cannot be obviously mapped to neural spiking data [47].

(2) **Compactness**, i.e. among models with similar scores, we prefer simpler models as they are potentially easier to understand and more efficient to experiment with. However, there are many ways to define this simplicity. Motivated by the observation that the feedforward path from retinal input to IT is fairly limited in length (e.g., [43]), for the purposes of this study we use **depth** as a simple proxy to meeting the biological constraint in artificial neural networks. Here we defined depth as the number of convolutional and fully connected layers in the longest feedforward path of a model.

(3) **Recurrence**: while core object recognition was originally believed to be largely feedforward because of its fast time scale [7], it has long been suspected that recurrent connections must be relevant for some aspects of object perception [22, 1, 44], and recent studies have shown their role even at short time scales [16, 42, 35, 32, 5]. Moreover, responses in the visual system have a temporal profile, so models at least should be able to produce responses *over time* too.

### 2.1 CORnet-S model specifics

CORnet-S (Fig. 1) aims to rival the best models on Brain-Score by transforming very deep feedforward architectures into a shallow recurrent model. Specifically, CORnet-S draws inspiration from ResNets that are some of the best models on our behavioral benchmark (Fig. 1; [33]) and can be thought of as unrolled recurrent networks [25]. Recent studies further demonstrated that weight sharing in ResNets was indeed possible without a significant loss in CIFAR and ImageNet performance [15, 23].

Moreover, CORnet-S specifically commits to an anatomical mapping to brain areas. While for comparison models we establish this mapping by searching for the layer in the model that best explains responses in a given brain area, ideally such mapping would already be provided by the model, leaving no free parameters. Thus, CORnet-S has four computational areas, conceptualized as analogous to the ventral visual areas V1, V2, V4, and IT, and a linear category decoder that maps from the population of neurons in the model's last visual area to its behavioral choices. This simplistic assumption of clearly separate regions with repeated circuitry was a first step for us to aim at building as shallow a model as possible, and we are excited about exploring less constrained mappings (such as just treating everything as a neuron without the distinction into regions) and more diverse circuitry (that might in turn improve model scores) in the future.

Each visual area implements a particular neural circuitry with neurons performing simple canonical computations: convolution, addition, nonlinearity, response normalization or pooling over a receptive field. The circuitry is identical in each of its visual areas (except for $V1_{COR}$), but we vary the total number of neurons in each area. Due to high computational demands, first area $V1_{COR}$ performs a $7 \times 7$ convolution with stride 2, $3 \times 3$ max pooling with stride 2, and a $3 \times 3$ convolution. Areas $V2_{COR}$, $V4_{COR}$ and $IT_{COR}$ perform two $1 \times 1$ convolutions, a bottleneck-style $3 \times 3$ convolution with stride 2, expanding the number of features fourfold, and a $1 \times 1$ convolution. To implement recurrence, outputs of an area are passed through that area several times. For instance, after $V2_{COR}$ processed the input once, that result is passed into $V2_{COR}$ again and treated as a new input (while the original input is discarded, see "gate" in Fig. 1). $V2_{COR}$ and $IT_{COR}$ are repeated twice, $V4_{COR}$ is repeated four times as this results in the most minimal configuration that produced the best model as determined by our scores (see Fig. 4). As in ResNet, each convolution (except the first $1 \times 1$) is followed by batch normalization [14] and ReLU nonlinearity. Batch normalization was not shared over time as suggested by Jastrzebski et al. [15]. There are no across-area bypass or across-area

feedback connections in the current definition of CORnet-S and retinal and LGN processing are not explicitly modeled.

The decoder part of a model implements a simple linear classifier – a set of weighted linear sums with one sum for each object category. To reduce the amount of neural responses projecting to this classifier, we first average responses over the entire receptive field per feature map.

## 2.2 Implementation Details

We used PyTorch 0.4.1 and trained the model using ImageNet 2012 [34]. Images were preprocessed (1) for training – random crop to $224 \times 224$ pixels and random flipping left and right; (2) for validation - central crop to $224 \times 224$ pixels; (3) for Brain-Score – resizing to $224 \times 224$ pixels. In all cases, this preprocessing was followed by normalization by mean subtraction and division by standard deviation of the dataset. We used a batch size of 256 images and trained on 2 GPUs (NVIDIA Titan X / GeForce 1080Ti) for 43 epochs. We use similar learning rate scheduling to ResNet with more variable learning rate updates (primarily in order to train faster): 0.1, divided by 10 every 20 epochs. For optimization, we use Stochastic Gradient Descent with momentum .9, a cross-entropy loss between image labels and model predictions (logits).

ImageNet-pretrained CORnet-S is available at github.com/dicarlolab/cornet.

## 2.3 Comparison to other models

Liang & Hu [24] introduced a deep recurrent neural network intended for object recognition by adding a variant of a simple recurrent cell to a shallow five-layer convolutional neural network backbone. Zamir et al. [49] built a more powerful version by employing LSTM cells, and a similar approach was used by [38] who showed that a simple version of a recurrent net can improve network performance on an MNIST-based task. Liao & Poggio [25] argued that ResNets can be thought of as recurrent neural networks unrolled over time with non-shared weights, and demonstrated the first working version of a folded ResNet, also explored by [15].

However, all of these networks were only tested on CIFAR-100 at best. As noted by Nayebi et al. [29], while many networks may do well on a simpler task, they may differentiate once the task becomes sufficiently difficult. Moreover, our preliminary testing indicated that non-ImageNet-trained models do not appear to score high on Brain-Score, so even for practical purposes we needed models that could be trained on ImageNet. Leroux et al. [23] proposed probably the first recurrent architecture that performed well on ImageNet. In an attempt to explore the recurrent net space in a more principled way, Nayebi et al. [29] performed a large-scale search in the LSTM-based recurrent cell space by allowing the search to find the optimal combination of local and long-range recurrent connections. The best model demonstrated a strong ImageNet performance while being shallower than feedforward controls. In this work, we wanted to go one step further and build a maximally compact model that would nonetheless yield top Brain-Score and outperform other recurrent networks on ImageNet.

# 3 Brain-Score: Comparing models to brain

To obtain quantified scores for brain-likeness, we built *Brain-Score*, a composite benchmark that measures how well models can predict (a) mean neural response of each neural recording site to each and every tested naturalistic image in non-human primate visual areas V4 and IT (data from [28]); (b) mean pooled human choices when reporting a target object to each tested naturalistic image (data from [33]), and (c) when object category is resolved in non-human primate area IT (data from [16]). To rank models on an overall score, we take the mean of the behavioral score, the V4 neural score, the IT neural score, and the neural dynamics score (explained below).

Brain-Score is open-sourced as a platform to score neural networks on brain data through the Brain-Score.org website for an overview of scores and through github.com/brain-score.

## 3.1 Neural predictivity

Neural predictivity is used to evaluate how well responses to given images in a source system (e.g., a deep ANN) predict the responses in a target system (e.g., a single neuron's response in visual area IT;

[48]). As inputs, this metric requires two assemblies of the form stimuli $\times$ neuroid where neuroids can either be neural recordings or model activations.

A total of 2,560 images containing a single object pasted randomly on a natural background were presented centrally to passively fixated monkeys for 100 ms and neural responses were obtained from 88 V4 sites and 168 IT sites. For our analyses, we used normalized time-averaged neural responses in the 70-170 ms window. For models, we reported the most predictive layer or (for CORnet-S) designated model areas and the best time point.

Source neuroids were mapped to each target neuroid linearly using a PLS regression model with 25 components. The mapping procedure was performed for each neuron using 90% of image responses and tested on the remaining 10% in a 10-fold cross-validation strategy with stratification over objects. In each run, the weights were fit to map from source neuroids to a target neuroid using training images, and then using these weights predicted responses were obtained for the held-out images. To speed up this procedure, we first reduced input dimensionality to 1000 components using PCA. We used the neuroids from V4 and IT separately to compute these fits. The median over neurons of the Pearson's $r$ between the predicted and actual response constituted the final neural fit score for each visual area.

## 3.2 Behavioral predictivity

The purpose of behavioral benchmarks it to compute the similarity between source (e.g., an ANN model) and target (e.g., human or monkey) behavioral responses in any given task [33]. For core object recognition tasks, primates (both human and monkey) exhibit behavioral patterns that differ from ground truth labels. Thus, our primary benchmark here is a behavioral response pattern metric, not an overall accuracy metric, and higher scores are obtained by ANNs that produce and predict the primate patterns of successes and failures. One consequence of this is that ANNs that achieve 100% accuracy will not achieve a perfect behavioral similarity score.

A total of 2,400 images containing a single object pasted randomly on a natural background were presented to 1,472 humans for 100 ms and they were asked to choose from two options which object they saw. For further analyses, we used participants response accuracies of 240 images that had around 60 responses per object-distractor pair (~300,000 unique responses). For evaluating models, we used model responses to 2,400 images from the layer just prior to 1,000-value category vectors. 2,160 of those images were used to build a 24-way logistic regression decoder, where each 24-value vector entry is the probability that a given object is in the image. This regression was then used to estimate probabilities for the 240 held-out images.

Next, both for human model responses, for each image, all normalized object-distractor pair probabilities were computed from the 24-way probability vector as follows: $\frac{p(\text{truth})}{p(\text{truth})+p(\text{choice})}$. These probabilities were converted into a $d'$ measure: $d' = Z(\text{Hit Rate}) - Z(\text{False Alarms Rate})$, where $Z$ is the estimated z-score of responses, Hit Rate is the accuracy of a given object-distractor pair, and the False Alarms Rate corresponds to how often the observers incorrectly reported seeing that target object in images where another object was presented. For instance, if a given image contained a dog and distractor was a bear, the Hit Rate for the dog-bear pair for that image came straight from the $240 \times 24$ matrix, while in order to obtain the False Alarms Rate, all cells from that matrix that did not have dogs in the image but had a dog as a distractor were averaged, and 1 minus that value was used as a False Alarm Rate. All $d'$ above 5 were clipped. This transformation helped to remove bias in responses and also to diminish ceiling effects (since many primate accuracies were close to 1), but empirically observed benefits of $d'$ in this dataset were small; see [33] for a thorough explanation. The resulting response matrix was further refined by subtracting the mean $d'$ across trials of the same object-distractor pair (e.g., for dog-bear trials, their mean was subtracted from each trial). Such normalization exposes variance unique to each image and removes global trends that may be easier for models to capture. The behavioral predictivity score was computed as a Pearson's $r$ correlation between the actual primate behavioral choices and model's predictions.

## 3.3 Object solution times

A total of 1318 grayscale images, containing images from Section 3.1 and MS COCO [26], were presented centrally to behaving monkeys for 100 ms and neural responses were obtained from 424 IT sites. Similar to [16], we fit a linear classifier on 90% of each 10 ms of model activations between

70-250 ms and used it to decode object category in each image from the non-overlapping 10% of the data. The linear classifier was based on a fully-connected layer followed by a softmax, with Xavier initialization for the weights [8], $l2$ regularized and decaying with $0.463$, inputs were z-scored, and fit with a cross-entropy loss, a learning rate of $1e^{-4}$ over 40 epochs with a training batch size of 64, and stopped early if the loss-value went below $1e^{-4}$. The predictions were converted to normalized $d'$ scores per image ("I1" in [33]) and per time bin. By linearly interpolating between these bins, we determined the exact millisecond when the prediction surpassed a threshold value defined by the monkey's behavioral output for that image, which we refer to as "object solution times", or OSTs. Images for which either the model or the neural recordings did not produce an OST because the behavioral threshold was not met were ignored. We report a Spearman correlation between the model OSTs and the actual monkey OSTs (as computed in [16]).

### 3.4 Generalization to new datasets

**Neural: New neurons, old images**  We evaluated models on an independently collected neural dataset (288 neurons, 2 monkeys, 63 trials per image; [16]) where new monkeys were presented with a subset of 640 images from the 2,560 images we used for neural predictivity.

**Neural: New neurons, new images**  We obtained a neural dataset from [16] for a selection of 1,600 of grayscale MS COCO images [26]. These images are very dissimilar from the synthetic images we used in other tests, providing a strong means to test Brain-Score generalization. The dataset consisted of 288 neurons from 2 monkeys and 45 trials per image. Unlike our previous datasets, this one had a low internal consistency between neural responses, presumably due to the electrodes being near their end of life and producing unreasonably high amounts of noise. We therefore only used the 86 neurons with internal consistency of at least 0.9.

**Behavioral: New images**  We collected a new behavioral dataset, consisting of 200 images (20 objects $\times$ 10 images) from Amazon Mechanical Turk users (185,106 trials in total). We used the same experimental paradigm as in our original behavioral test but none of the objects were from the same category as before.

**CIFAR-100**  Following the procedure described in [18], we tested how well these models generalize to CIFAR-100 dataset by only allowing a linear classifier to be retrained for the 100-way classification task (that is, without doing any fine-tuning). As in [18], we used a scikit-learn implementation of a multinomial logistic regression using L-BFGS [31], with the best C parameter found by searching a range from .0005 to .05 in 10 logarithmic steps (40,000 images from CIFAR-100 train set were used for training and the remaining 10,000 for testing; the search range was reduced from [18] because in our earlier tests we found that all models had their optimal parameters in this range). Accuracies reported on the 10,000 test images.

## 4 Results

### 4.1 CORnet-S is the best brain-predicting model so far

We performed a large-scale model comparison using most commonly used neural network families: AlexNet [19], VGG [37], ResNet [10], Inception [39–41], SqueezeNet [13], DenseNet [12], MobileNet [11], and (P)NASNet [50, 27]. These networks were taken from publicly available checkpoints: AlexNet, SqueezeNet, ResNet-{18,34} from PyTorch [30]; Inception, ResNet-{50,101,152}, (P)NASNet, MobileNet from TensorFlow-Slim [36]; and Xception, DenseNet, VGG from Keras [3]. As such, the training procedure is different between models and our results should be related to those model instantiations and not to architecture families. To further map out the space of possible architectures, we included a family of models called *BaseNets*: lightweight AlexNet-like architectures with six convolutional layers and a single fully-connected layer, captured at various stages of training. Various hyperparameters were varied between BaseNets, such as kernel sizes, nonlinearities, learning rate etc.

Figure 1 shows how models perform on Brain-Score and ImageNet. CORnet-S outperforms other alternatives by a large margin with the Brain-Score of .471. Top ImageNet models also perform well, with leading models stemming from the DenseNet and ResNet families.Interestingly, models

that rank the highest on ImageNet performance are also not the ones scoring high on brain data, suggesting a potential disconnect between ImageNet performance and fidelity to brain mechanisms. For instance, despite its superior performance of 82.9% top-1 accuracy on ImageNet, PNASNet only ranks 13[th] on the overall Brain-Score. Models with an ImageNet top-1 performance below 70% show a strong correlation with Brain-Score of .90 but above 70% ImageNet performance there was no significant correlation ($p \gg .05$, cf. Figure 1).

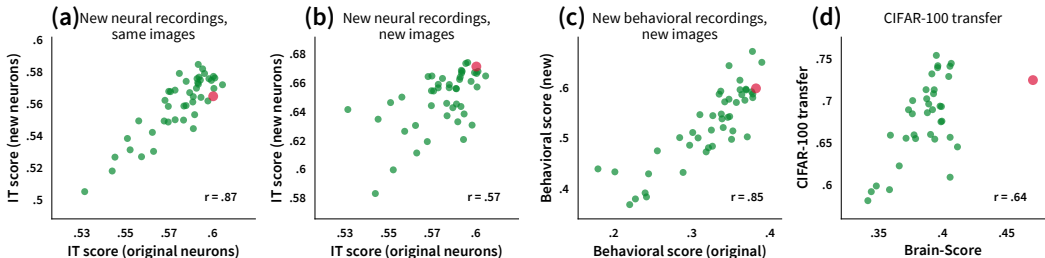

Figure 2: **Brain-Score generalization across datasets: (a)** to neural recordings in new subjects with the same stimulus set, **(b)** to neural recordings in new subjects with a very different stimulus set (MS COCO), **(c)** to behavioral responses in new subjects with new object categories, **(d)** to CIFAR-100.

We further asked if Brain-Score reflects idiosyncrasies of the particular datasets that we included in this benchmark or instead, more desirably, provides an overall evaluation of how brain-like models are. To address this question, we performed four different tests with various generalization demands (Fig. 2; CORnet-S was excluded). First, we compared the scores of models predicting IT neural responses to a set of new IT neural recordings [16] where new monkeys were shown the same images as before. We observed a strong correlation between the two sets (Pearson $r = .87$). When compared on predicting IT responses to a very different image set (1600 MS COCO images [26]), model rankings were still strongly correlated (Pearson $r = .57$). We also found a strong correlation between model scores on our original behavioral set and a newly obtained set of behavioral responses to images from 20 new categories that were not used before (200 images total; Pearson $r = .85$). Finally, we evaluated model feature generalization to CIFAR-100 without fine-tuning (following Kornblith et al. [18]). Again, we observed a compelling correlation to Brain-Score values (Pearson $r = .64$). Overall, we expect that adding more benchmarks to Brain-Score will further lead scores to converge.

## 4.2 CORnet-S is the best on ImageNet and CIFAR-100 among shallow models

Due to anatomical constraints imposed by the brain, CORnet-S's architecture is much more compact than the majority of deep models in computer vision (Fig. 3 middle). Compared to similar models with a depth less than 50, CORnet-S is shallower yet better than other models on ImageNet top-1 classification accuracy. AlexNet and IamNN are even shallower (depth of 8 and 14) but suffer on classification accuracy (57.7% and 69.6% top-1 respectively) – CORnet-S provides a good trade-off between the two with a depth of 15 and top-1 accuracy of 73.1%. Several epochs later in training top-1 accuracy actually climbed to 74.4% but since we are optimizing for the brain, we chose the epoch with maximum Brain-Score. CORnet-S also achieves the best transfer performance among similarly shallow models (Fig. 3, right), indicating the robustness of this model.

## 4.3 CORnet-S mediates between compactness and high performance through recurrence

To determine which elements in the circuitry are critical to CORnet-S, we attempted to alter its block structure and record changes in Brain-Score (Fig. 4). We only used V4, IT, and behavioral predictivity in this analysis in order to understand the non-temporal value of CORnet-S structure. We found that the most important factor was the presence of at least a few steps of recurrence in each block. Having a fairly wide bottleneck (at least 4x expansion) and a skip connection were other important factors. On the other hand, adding more recurrence or having five areas in the model instead of four did not improve the model or hurt its Brain-Score. Other factors affected mostly ImageNet performance, including using two convolutions instead of three within a block, having more areas in the model and using batch normalization per time step instead of a global group normalization [45]. The type of gating did not seem to matter. However, note that we kept training with identical hyperparameters

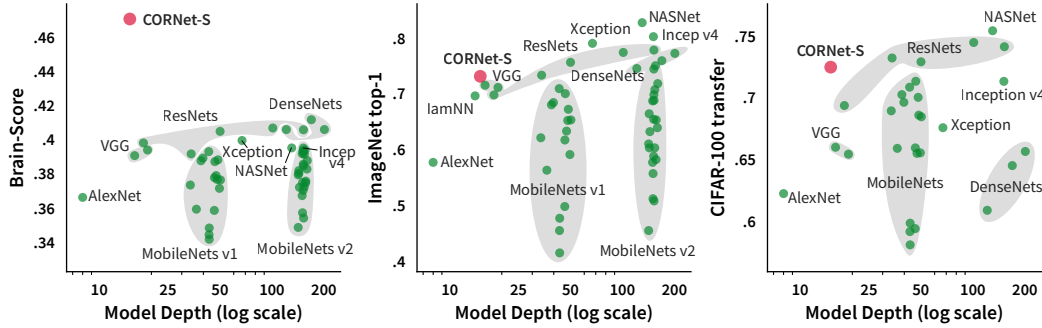

Figure 3: **Depth versus (left) Brain-Score, (middle) ImageNet top-1 performance, and (right) CIFAR-100 transfer performance.** Most simple models perform poorly on Brain-Score and ImageNet, and generalize less well to CIFAR-100, while the best models are very deep. CORnet-S offers the best of both worlds with the best Brain-Score, compelling ImageNet performance, the shallowest architecture we could achieve to date, and the best transfer performance to CIFAR-100 among shallow models. (Note: dots corresponding to MobileNets were slightly jittered along the x-axis to improve visibility.)

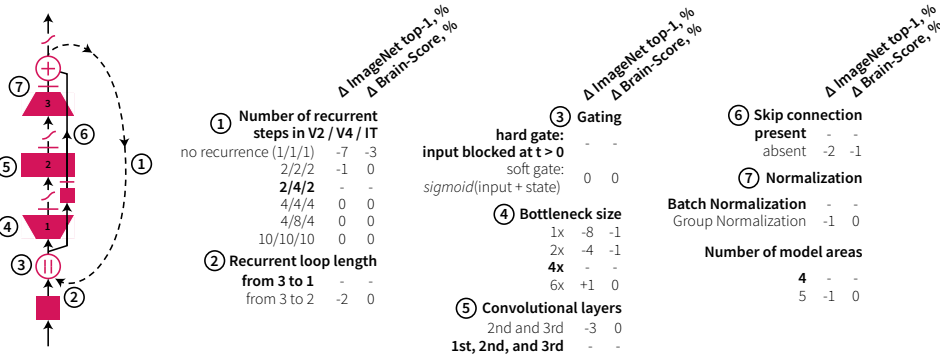

Figure 4: **CORnet-S circuitry analysis.** Each row indicates how ImageNet top-1 and Brain-Score change with respect to the baseline model (in bold) when a particular hyperparameter is changed. The OSTs of IT are not included in the Brain-Score here.

for all these model variants. We therefore cannot rule out that the reported differences could be minimized if more optimal hyperparameters were found.

## 4.4 CORnet-S captures neural dynamics in primate IT

Feed-forward networks cannot make any dynamic predictions over time, and thus cannot capture a critical property of the primate visual system [16, 42]. By introducing recurrence, CORnet-S is capable of producing temporally-varying response trajectories in the *same set of neurons*. Recent experimental results [16] reveal that the linearly decodable solutions to object recognition are not all produced at the same time in the IT neural population – images that are particularly challenging for deep ANNs take longer to evolve in IT. This timing provides a strong test for the model: Does it predict image-by-image temporal trajectories in IT neural responses over time? We thus estimated for each image when explicit object category information becomes available in CORnet-S – termed "object solution time" (OST) – and compared it with the same measurements obtained from monkey IT cortex [16]. Importantly, the model was never trained to predict monkey OSTs. Rather, a linear classifier was trained to decode object category from neural responses and from model's responses at each 10 ms window (Section 3.3). OST is defined as the time when this decoding accuracy reaches a threshold defined by monkey behavioral accuracy. We converted the two IT timesteps in CORnet-S to milliseconds by setting $t_0 \hat{=}$ 0-150 ms and $t_1 \hat{=}$ 150+ ms. We evaluated how well CORnet-S could capture the fine-grained temporal dynamics in primate IT cortex and report a correlation score of .25 ($p < 10^{-6}$; Figure 5). Feed-forward models cannot capture neural dynamics and thus scored 0.

## 5 Discussion

We developed a relatively shallow recurrent model CORnet-S that follows neuroanatomy more closely than standard machine learning ANNs, and is among the top models on Brain-Score yet remains competitive on ImageNet and on transfer tests to CIFAR-100. As such, it combines the best of both neuroscience desiderata and machine learning engineering requirements, demonstrating that models that satisfy both communities can be developed.

While we believe that CORnet-S is a closer approximation to the anatomy of the ventral visual stream than current state-of-the-art deep ANNs because we specifically limit the number of areas and include recurrence, it is still far from complete in many ways. From a neuroscientist's point of view, on top of the lack of biologically-plausible learning mechanisms (self-supervised or unsupervised), a better model of the ventral visual pathway would include more anatomical and circuitry-level details, such as retina or lateral geniculate nucleus. Similarly, adding a skip connection was not informed by cortical circuitry properties but rather proposed by He et al. [10] as a means to alleviate the degradation problem in very deep architectures. But we note that not just any architectural choices work. We have tested hundreds of architectures before finding CORnet-S type of circuitries (Figure 4).

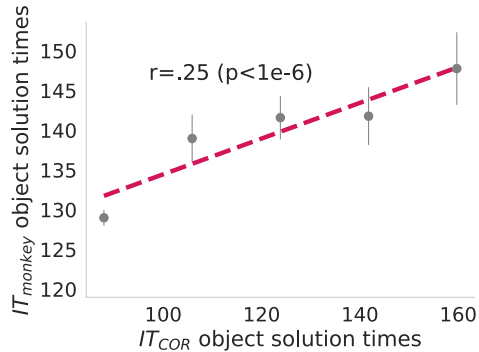

Figure 5: **CORnet-S captures neural dynamics.** A linear decoder is fit to predict object category at each 10 ms window of IT responses in model and monkey. We then tested object solution times (OST) per image, i.e. when $d'$ scores of model and monkey surpass the threshold of monkey behavioral response. $r$ is computed on the raw data, whereas the plot visualizes binned OSTs. Error bars denote s.e.m. across images.

A critical component in establishing that models such as CORnet-S are strong candidate models for the brain is Brain-Score, a framework for quantitatively comparing any artificial neural network to the brain's neural network for visual processing. Even with the relatively few brain benchmarks that we have included so far, the framework already reveals interesting patterns. First, it extends prior work showing that performance correlates with brain similarity. However, adding recurrence allows us to break from this trend and achieve much better alignment to the brain. Even when the OST measure is not included in Brain-Score, CORnet-S remains one of the top models, indicating its general utility. On the other hand, we also find a potential disconnect between ImageNet performance and Brain-Score with PNASNet, a state-of-the-art model on ImageNet used in our comparisons, that is not performing well on brain measures, whereas even small networks with poor ImageNet performance achieve reasonable scores. We further observed that models that score high on Brain-Score also tend to score high on other datasets, supporting the idea that Brain-Score reflects how good a model is overall, not just on the four particular neural and behavioral benchmarks that we used.

However, it is possible that the observed lack of correlation is only specific to the way models were trained, as reported recently by Kornblith et al. [18]. For instance, they found that the presence of auxiliary classifiers or label smoothing does not affect ImageNet performance too much but significantly decreases transfer performance, in particular affecting Inception and NASNet family of models, i.e., the ones that performed worse on Brain-Score than their ImageNet performance would imply. Kornblith et al. [18] reported that retraining these models with optimal settings markedly improved transfer accuracy. Since Brain-Score is also a transfer learning task, we cannot rule out that Brain-Score might change if we retrained the affected models classes. Thus, we reserve our claims only about the specific pre-trained models rather than the whole architecture classes.

More broadly, we suggest that models of brain processing are a promising opportunity for collaboration between neuroscience and machine learning. These models ought to be compared through quantified scores on how brain-like they are, which we here evaluate with a composite of many neural and behavioral benchmarks in Brain-Score. With CORnet-S, we showed that neuroanatomical alignment to the brain in terms of compactness and recurrence can better capture brain processing by predicting neural firing rates, image-by-image behavior, and even neural dynamics, while simultaneously maintaining high ImageNet performance and outperforming similarly compact models.

**Acknowledgments**

We thank Simon Kornblith for helping to conduct transfer tests to CIFAR, and Maryann Rui and Harry Bleyan for the initial prototyping of the CORnet family.

This project has received funding from the European Union's Horizon 2020 research and innovation programme under grant agreement No 705498 (J.K.), US National Eye Institute (R01-EY014970, J.J.D.), Office of Naval Research (MURI-114407, J.J.D), the Simons Foundation (SCGB [325500, 542965], J.J.D; 543061, D.L.K.Y), the James S. McDonnell foundation (220020469, D.L.K.Y.) and the US National Science Foundation (iis-ri1703161, D.L.K.Y.). This work was also supported in part by the Semiconductor Research Corporation (SRC) and DARPA. The computational resources and services used in this work were provided in part by the VSC (Flemish Supercomputer Center), funded by the Research Foundation - Flanders (FWO) and the Flemish Government – department EWI.

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
