[Supplementary Material]

# A   Numerical Brain-Scores

Table 1: Brain-Scores and individual performances for state-of-the-art models

| Model | Brain-Score | V4 | IT | OST | Behavior |
|---|---|---|---|---|---|
| CORnet-S | **.471** | .65 | .6 | **.25** | .382 |
| DenseNet-169 | .412 | .663 | **.606** | 0 | .378 |
| ResNet-101 v2 | .407 | .653 | .585 | 0 | **.389** |
| DenseNet-201 | .406 | .655 | .601 | 0 | .368 |
| DenseNet-121 | .406 | .657 | .597 | 0 | .369 |
| ResNet-152 v2 | .406 | .658 | .589 | 0 | .377 |
| ResNet-50 v2 | .405 | .653 | .589 | 0 | .377 |
| Xception | .399 | .671 | .565 | 0 | .361 |
| Inception v2 | .399 | .646 | .593 | 0 | .357 |
| Inception v1 | .399 | .649 | .583 | 0 | .362 |
| ResNet-18 | .398 | .645 | .583 | 0 | .364 |
| NASnet Mobile | .398 | .65 | .598 | 0 | .342 |
| PNASnet Large | .396 | .644 | .59 | 0 | .351 |
| Inception ResNet v2 | .396 | .639 | .593 | 0 | .352 |
| NASnet Large | .395 | .65 | .591 | 0 | .339 |
| Best MobileNet | .395 | .613 | .59 | 0 | .377 |
| VGG-19 | .394 | **.672** | .566 | 0 | .338 |
| Inception V4 | .393 | .628 | .575 | 0 | .371 |
| Inception V3 | .392 | .646 | .587 | 0 | .335 |
| ResNet-34 | .392 | .629 | .559 | 0 | .378 |
| VGG-16 | .391 | .669 | .572 | 0 | .321 |
| Best BaseNet | .378 | .663 | .594 | 0 | .256 |
| AlexNet | .366 | .631 | .589 | 0 | .245 |
| SqueezeNet v1.1 | .351 | .652 | .553 | 0 | .201 |
| SqueezeNet v1.0 | .341 | .641 | .542 | 0 | .18 |

# B Brain-Score benchmark details

## B.1 Brain-Score benchmark

To evaluate how well a model is doing overall, we computed the global Brain-Score as a composite of neural V4 predictivity score, neural IT predictivity score, object solution times in IT, and behavioral I2n predictivity score (each of these scores was computed as described in main text). The Brain-Score presented here is the mean of the four scores. This approach of taking the mean does not normalize by different scales of the scores so it may be penalizing scores with low variance. However, the alternative approach of ranking models on each benchmark separately and then taking the mean rank would impose the strong assumption that for any two models with (even insignificantly) different scores, their ranks are also different. We thus chose to take the mean score to preserve the distance in values.

## B.2 Neural recordings

The neural dataset currently used in both neural benchmarks included in this version of Brain-Score is comprised of neural responses to 2,560 naturalistic stimuli in 88 V4 neurons and 168 IT neurons, collected by [1]. The image set consists of 2,560 grayscale images in eight object categories (animals, boats, cars, chairs, faces, fruits, planes, tables). Each category contains eight unique objects (for instance, the "face" category has eight unique faces). The image set was generated by pasting a 3D object model on a naturalist background. In each image, the position, pose, and size of an object was randomly selected in order to create a challenging object recognition task both for primates and machines. A circular mask was applied to each image (see [1] for details on image generation).

Two macaque monkeys were implanted three arrays each, with one array placed in area V4 and the other two placed on the posterior-anterior axis of IT cortex. The monkeys passively observed a series of images (100 ms image duration with 100 ms of gap between each image) that each subtended approximately 8 deg visual angle. To obtain a stable estimate of the neural responses to each image, each each image was re-tested about 50 times (re-tests were randomly interleaved with other images). In the benchmarks used here, we used an average neural firing rate (normalized to a blank gray image response) in the window between 70 ms and 170 ms after image onset where the majority of object category-relevant information is contained [1].

## B.3 Behavioral recordings

The behavioral data used in the current round of benchmarks was obtained by [3] and [4]. Here we used only the human behavioral data, but the human and non-human primate behavioral patterns are very similar to each other [3, 4]. The image set used in this data collection was generated in a similar way as the images for V4 and IT using 24 object categories, and human responses were collected on Amazon Mechanical Turk. (Other details are explained in the main text.)

## C  Depth

From a neuroscience point of view, simpler models can be better mapped to cortex and be better analyzed and understood with regard to the brain. Simpler models can also be better made sense of sense in terms of what components constitute a strong model by reducing models to their most essential elements. One possibility was to use the total number of parameters (weights). However, it did not seem to map well to simplicity in neuroscience terms. For instance, a single convolutional layer with many filter maps could have many parameters yet it seems much simpler than a multilayer branching structure, like the Inception block [5], that may have less parameters overall.

Moreover, our models are always tested on independent data sampled from different distributions than the train data. Thus, after training a model, all these parameters were fixed for the purposes of brain benchmarks, and the only free parameters are the ones introduced by the linear decoder that is trained on top of the frozen model's parameters (see above for decoder details).

We also considered computing the total number of convolutional and fully-connected layers, but some models, like Inception, perform some convolutions in parallel, while others, like ResNeXt [6], group multiple convolutions into a single computation. We thus decided to use the "longest path" definition as described in the main text.

## D  Predictors of neural scores

We compared model scores on neural (V4, IT) recordings with the scores on behavioral recordings to see if e.g. a behavioral benchmark alone would already be sufficient or if the entire set of benchmarks is necessary. We found that there was a correlation to behavior (.65 for V4 and .87 or IT) which is strong enough to connect neurons to behavior but not sufficient for behavior alone to explain the entire neural population, warranting a composite set of benchmarks.

Moreover, we tested if the number of features in model layers might predict the neural scores. Even though we PCA all features to 1,000 components, higher dimensionality might result in better scores. Following Figure 1, we found that neural scores are not consistently correlated with the number of features across neural benchmarks: for V4, having more than 1000 features helps a little ($r = .46, p < .05$) but for IT, there was no significant correlation at any number of features.

Figure 1: **Neural Scores do not depend on number of features.** We plot the number of features in models' highest-scoring layers against their neural (V4 and IT) scores. The number of neurons does not appear to be a predictor of better brain-likeness.

# E  Early and late neural predictions

Focusing on the temporal aspect of our neural data, we divided spike rates into an early time bin ranging from 90-110 ms and a late time bin from 190-210 ms. We found that this early-late division highlighted functional model difference more prominently than the mean temporal prediction in [2]. For instance, Figure 2 shows how IT is predicted well by strong ImageNet models at a late stage, but not at early stages. CORnet-S does well on both of these predictions.

Figure 2: **Prediction correlations on early and late spike rates.** We compare ImageNet performance against pearson correlation of predicted spike rates with neural data binned into early (90-110 ms) and late (190-210 ms). Model mappings are performed separately per bin, layers are chosen based on 70-170 ms scores. Notice how better ImageNet models are better at predicting late IT responses, but not early ones.

# F   CORnet-S search

Figure 3: **ImageNet top-1 performance vs. behavioral benchmark on various CORnets.** We manually tried many different configurations of CORnet circuitry. The figure is showing how behavioral benchmark of Brain-Score is related to ImageNet top-1 performance in 106 CORnet configurations. Each dot corresponds to a particular CORnet at a particular point during training. The correlation between ImageNet top-1 performance and CORnet is robust but there is also high variance in this relationship. In particular, notice how some models achieve close to 75% ImageNet performance but show only a mediocre behavioral score. Thus, optimizing solely for ImageNet is not guaranteed at all to lead to a good alignment to brain data.