[Reviews · NeurIPS 2019]

Reviewer 1



Although the idea of a functional correspondence between ANN components and brain regions is a key motivating idea in this paper, it has to be noted that brain “areas” can be functionally and anatomically heterogeneous. Therefore, the one-to-one mapping between the number of model components and the number of brain regions may be a bit arbitrary and simplistic. Can we really say for sure it should be four areas, and not five or six? Moreover, the assumption that the circuitry does not differ across regions seems simplistic. Lines 89-101: How are these architecture details decided on? For example, “V2_COR and IT_COR are repeated twice, V4_COR is repeated four times”. As far as I can see, there is no a priori justification or explanation for many of these choices. If there has been extensive preliminary architecture search then this should be stated clearly. (The circuitry analysis (Fig 5) does mitigate this point somewhat). Although the paper is generally well-written, there are a number of places where the text doesn’t really flow (perhaps as a result of extensive editing and re-organisation). For example, “Feedforward Simplicity” is mentioned on line 59 but without any definition or explanation. (Personally, I find the phrase a bit strange, since the whole point of the models is that they are not feedforward - - perhaps “information flow simplicity” or just “model simplicity” is better?) Line 164: counting recurrent paths only once seems a little strange: shouldn’t a model with recurrence be measured as more complex than one without recurrence, all else being equal? Line 206: I am not sure it is clear what “category” means here. Line 253: I realise space is a limiting factor, but I think it would be good to have more detailed information about the object recognition task the monkey is performing. The evolution of representations in recurrent neural network models of object vision over time and their relation to neural data over time (e.g. line 76) is tested in Clarke et al 2018 (JoCN), in relation to human MEG data. One general comment I have about this paper is that it is not always clear what this paper’s novel contribution is, in relation to other pre-prints published by what I suspect is the same lab. There are a number of interrelated arXiv preprints on this work, and it is not always clear what is novel in each. For example, is this current paper to be taken as the first published presentation of the “Brain-Score” benchmark? If so, it should be described much more extensively than it is. If not, it should not be regarded as a novel contribution of this paper (just reference the earlier preprint).

Reviewer 2



UPDATE AFTER AUTHOR RESPONSE Thank you for the clarifications. I had only one real reservation, regarding the rationale for feedforward simplicity, and this has been clarified. It was a pleasure to review this. ------------------------------ Summary: The paper develops a multi-faceted metric that summarizes how similar a deep network's representations and decisions are to those of the primate ventral visual stream. This builds on a body of recent work that has identified parallels between deep networks and the ventral stream. The past work suggested that deep networks are good but incomplete models of the ventral stream, hinting that better models (and thus richer understanding) might be achieved by somehow modifying deep networks. To make clear progress in this direction, an objective and reasonably comprehensive measure of brain similarity is needed, and this submission provides it. The authors also present a new network model that establishes a new state of the art for this metric. In contrast with established deep networks (some of which score fairly well) this one has structural parallels with the ventral stream, including analogous areas and recurrent connections. The authors also show that while object recognition performance is correlated with brain similarity, this correlation is weak for the recent best-performing networks in the deep learning literature, suggesting a divergence from the brain, which the new metric provides a way to avoid. Originality: Each of the above contributions is highly original. Quality: This is thorough, field-leading work. In addition to the main contributions, the authors showed that the results generalized well to new images and monkeys (Fig. 2), and reported the effects of numerous variations on the model (Fig. 5). Clarity: The text is well written and the figures and nicely done. The supplementary material provides rich additional detail. A minor limitation is that, since the paper covers a lot of ground, some of the details go by quickly. For example, the description of the network might be expanded slightly. I am not sure the small square labelled "conv / stride 2" in Fig. 1 was explained in the text, or the kind of gating (except in Fig. 5). Significance: Each of the above contributions is highly significant and likely to inform other researchers' future work in this area.

Reviewer 3



In this manuscript, the authors design a brain-inspired convolutional recurrent neural network- CORnet-S that maps the visual ventral pathway to different layers in the network. The CORnet-S model achieves competitive accuracy on ImageNet; more importantly, it achieves state-of-the-art performance on the Brain-Score, a comprehensive benchmark to assess the performance of a model for neural predictivity, behavioral predictivity, object solution times (OST), and feedforward simplicity. The manuscript is clearly written, and is of interest to a broad audience in the conference. However, I do have several concerns for the manuscript: * The biggest concern is the justification of contribution. First, Brain-Score is proposed in 2018 [1]. Although [1] is a preprint version, but it has been widely cited, so I would argue the contribution of Brain-Score for this manuscript. For me, a novel contribution is the proposal of the OST metric that expands the Bran-Score benchmark from single frame to sequence level. Second, as can be seen in Table 1 in the supplemental material, the claimed state-of-the-art performance largely depends on the OST score, which is unfair to all other models without recurrent connections (as their scores are by default 0). If OST score is not taken into consideration, the performance of CORnet-S is not state-of-the-art anymore, although still competitive. In this case, in order to better justify the contribution, evidence from other perspectives are needed, such as the number of parameters, inference speed, GPU memory usage, etc. Another way the author may consider is to add recurrent connection to other feedforward models with similar depth/number of parameters. If CORnet-S is more efficient than deeper models or performs better than recurrent shallow models, then the contribution can still be justified. At the current stage, however, more experiments/metrics are needed. * Since OST is a novel contribution of the manuscript, section 4.6 and Figure 6 should be elaborated more clear. For example, why do we need a 80%-10%-10% setting for the CORnet-S setting? How does the 10ms window map to the dots in Figure 6 and how are they measured in the network? Minor concerns: * Figure 2: What does the red dot mean? * Figure 3: cannot see the superiority of the CORnet-S model. Since the performances are really close, this Figure may not be included. [1] Schrimpf, M., Kubilius, J., Hong, H., Majaj, N. J., Rajalingham, R., Issa, E. B., ... & Yamins, D. L. (2018). Brain-Score: which artificial neural network for object recognition is most brain-like?. BioRxiv, 407007. ===========Post-rebuttal============ My concern of the preprint has been addressed by clarification of policy by the AC, so is no longer an issue. As a result the Brain-Score benchmark has become a major contribution. The authors also addressed my concern regarding comparison against other shallow recurrent models (in perfomance, although I am still interested in memory usage/model size, etc). As a result my score is updated from 5 to 8.

[Author Response · NeurIPS 2019]

We would like to thank all reviewers for their comments and helpful feedback.

**Feedforward Simplicity.** R1 and R2 questioned Feedforward Simplicity as a measure as it is unclear what constitutes
a simpler model and that the brain is complex, so it is not exactly clear why simpler models would be preferred.
However, studies of response latencies and sequential processing in cortical areas demonstrate that the feedforward path
from retinal input to IT should be limited in length (e.g., see Tovée, Current Biology, 1994). Counting the number of
layers provides a simple proxy to meeting this biological constraint in artificial neural networks, and our Feedforward
Simplicity was meant to quantify this. However, we agree with the reviewers that this term is confusing and thus in the
revised version we will update it to simply "Depth" and clarify our reasoning as above.

**One-to-one mapping between brain areas and model components.** R1 commented that a one-to-one correspondence
between model components and hetereogeneous brain regions seems simplistic and that circuitry might not be the exact
same across regions. We agree with both of those points. The simplistic assumption of clearly separate regions with
repeated circuitry was a first step for us to aim at building as shallow a model as possible, and we are excited about
exploring less constrained mappings (such as just treating everything as a neuron without the distinction into regions)
and more diverse circuitry (that might in turn improve model scores) in the future.

**Justification for CORnet-S architectural choices.** R1 asked how the number of recurrent steps as well as other
architectural choices were justified. As R1 correctly pointed out, the major justification came from the ablation study in
Fig. 5 – these steps were the most minimal configuration that produced the best model as determined by our scores.
Training for more recurrent steps is possible, but at least on our current set of scores, we see no improvement. We
expect that future temporal benchmarks might warrant the need for more recurrent steps.

**Details sometimes lacking.** All reviewers noted that some details are lacking and, according to R1, if this is the first
publication of Brain-Score, many more details should be provided. The succinctness of the text is primarily due to the
hard limit of 8 pages, thus we placed many details in the Appendix. Following the reviewers' remarks, we will attempt
to work in the missing details and fixes for the camera-ready version; however, due to pages limits, we will still have to
rely on the Appendix for in-depth explanations.

To answer some of the details that the reviewers pointed out: (i) L206: category refers to the categories of images used
in new behavioral experiments; (ii) Fig. 1: conv / stride 2 refers to the stride-2 convolution; gating refers to a gate that
only lets information through at t=0 (though a soft (sigmoid) gate leads to similar results; Fig. 5); (iii) OST: 80/10/10
split was used to have independent train / validation / test sets. It was not tuned to CORnet-S.

**Brain-Score is not novel because it already available as a preprint.** We hope the reviewer R3 will reconsider this
criticism because, according to NeurIPS submission criteria, preprints are explicitly allowed, and this paper is the
first publication of Brain-Score. Moreover, this submission has also developed Brain-Score much further from the
preprint, namely, (i) we included four transfer tests on three newly collected datasets to validate the generalization
capabilities (Fig. 2); (ii) we investigated possible predictors of Brain-Score (Appendix B.3); and (iii) we developed a
mature open-source code base for an easy benchmarking.

**CORnet-S has an unfair advantage because of recurrent connections.**
R3 stated that CORnet-S is winning only because of OST predictions that
by design require recurrence. We agree with this remark and to a large
extent that is the point of this model: the widely used family of feed-forward
models simply cannot capture temporal processing that occurs in primate
ventral visual pathway and are thus insufficient to build brain-like models.
As also pointed out by R1, the value of CORnet-S does not lie in achiev-
ing the state-of-the-art on typical machine learning benchmarks but rather
in demonstrating that it is BOTH brain-like AND a competitive machine
learning model.

Figure 1: CORnet-S transfers better to CIFAR than similarly shallow networks

In addition, CORnet-S demonstrates competitive behavior on other machine learning measures: (a) it outperforms
comparably shallow feedforward models and shows the best transfer performance among similarly shallow models
(this response Figure 1); (b) it outperforms other shallow recurrent networks, as asked by R3 (see Appendix Fig. 2 that
plots many other variants of shallow recurrent models that we built and tested; while some achieve a higher ImageNet
performance, CORnet-S is the current best compromise between Brain-Score AND ImageNet performance).

Overall, we find that R3 evaluated this submission largely from a machine learning perspective. However, the strength
of this submission is that it addresses the expectations of BOTH machine learning and computational neuroscience
communities, unlike most prior work that was "either or". We are particularly excited about the impact this might have
on sparking discussions between these communities on building high-performing AND high-fidelity models of brain
function.

[Meta-Review · NeurIPS 2019]

There was a robust discussion on the merits and novelty of this work after the rebuttal from the authors. There was some confusion as to whether this work is truly the first to-be-published work that introduces BrainScore. We concluded that given the pre-print only versions of the work thus far, this qualifies as a genuinely new contribution (wrt. to the NeurIPS guidelines). With that in mind: the authors would do well do describe the details of the BrainScore in more detail in the camera ready version. Otherwise, everyone is excited about the novel aspects (architecture, experiments, metric etc) of this work so I wholeheartedly recommend this work to be accepted at NeurIPS.